# Insufficient yet improving involvement of the global south in top sustainability science publications

Olivier Dangles[1]*, Quentin Struelens[1], Mame-Penda Ba[2], Yvonne Bonzi-Coulibaly[3], Philippe Charvis[1], Evens Emmanuel[4], Carolina González Almario[5], Lahoucine Hanich[6,7], Ousmane Koita[8], Fabiola León-Velarde[9], Yvonne K. Mburu[10], Francine Ntoumi[11], Silvia Restrepo[12], Laurent Vidal[13]

1 Direction Déléguée à la Science, Institut de Recherche pour le Développement, Marseille, France,
2 Laboratoire d'Analyse des Sociétés et Pouvoirs/Afrique-Diaspora, Université Gaston Berger, Saint-Louis, Sénégal, 3 Département de Chimie, Université Joseph Ki-Zerbo, Ouagadougou, Burkina Faso, 4 Rectorat à la Recherche et à l'innovation, Université Quisqueya, Port-au-Prince, Haiti, 5 Departamento de Agrobiodiversidad, Corporación Colombiana de Investigación Agropecuaria AGROSAVIA, Mosquera, Cund., Colombia, 6 Laboratoire Géoressources-Département des Sciences de la Terre, Université Cadi Ayyad, Marrakech, Morocco, 7 Center for Remote Sensing Application, Mohamed VI Polytechnic University, Benguerir, Morocco, 8 Laboratoire de Biologie Moléculaire Appliquée, Université de Bamako, Bamako, Mali, 9 Departamento de Ciencias Biológicas y Fisiológicas, Universidad Peruana Cayetano Heredia, Lima, Perú, 10 Direction générale, Nexakili, Paris, France, 11 Faculté des Sciences et Techniques, Université Marien Ngouabi, Brazzaville, Congo, 12 Vicerrectoría de Investigación Creación, Universidad Los Andes, Bogotá, Colombia, 13 Représentation au Mali, Institut de Recherche pour le Développement, Bamako, Mali

* olivier.dangles@ird.fr

**Data Availability Statement:** The data underlying the results are within the paper and Supporting Information files.

## Abstract

The creation of global research partnerships is critical to produce shared knowledge for the implementation of the UN 2030 Agenda for Sustainable Development. Sustainability science promotes the coproduction of inter- and transdisciplinary knowledge, with the expectation that studies will be carried out through groups and truly collaborative networks. As a consequence, sustainability research, in particular that published in high impact journals, should lead the way in terms of ethical partnership in scientific collaboration. Here, we examined this issue through a quantitative analysis of the articles published in *Nature Sustainability* (300 papers by 2135 authors) and *Nature* (2994 papers by 46,817 authors) from January 2018 to February 2021. Focusing on these journals allowed us to test whether research published under the banner of sustainability science favoured a more equitable involvement of authors from countries belonging to different income categories, by using the journal *Nature* as a control. While the findings provide evidence of still insufficient involvement of Low-and-Low-Middle-Income-Countries (LLMICs) in *Nature Sustainability* publications, they also point to promising improvements in the involvement of such authors. Proportionally, there were 4.6 times more authors from LLMICs in *Nature Sustainability* than in *Nature* articles, and 68.8–100% of local Global South studies were conducted with host country scientists (reflecting the discouragement of parachute research practices), with local scientists participating in key research steps. We therefore provide evidence of the promising, yet still insufficient, involvement of low-income countries in top sustainability science publications and discuss ongoing initiatives to improve this.

**Funding:** The authors received no specific funding for this work.

**Competing interests:** The authors have declared that no competing interests exist.

## Introduction

Interconnected and intercultural scientific research that provides equal opportunities for every researcher represents a powerful way of moving towards sustainable development goals (SDGs) [1]. The UN 2030 Agenda for Sustainable Development stresses the need to revitalize transnational partnerships through enhancing "North–South, South–South and triangular regional and international cooperation" and "knowledge sharing on mutually agreed terms" (Target 17.6) [2]. While it is recognized that successful sustainability efforts require diversity, inclusion and equity [3], in practice, recent studies in a wide range of disciplines (including planetary health [4], biodiversity conservation [5], geoscience [6] and social sciences [7]) have shown pronounced asymmetry in the North–South relationship. In the worst cases, researchers from wealthier countries conduct research in Low-and-Lower-Middle-Income Countries (LLMICs) with little involvement of local researchers [8, 9]. Overall, while the SDGs have been widely officially endorsed by LLMIC governments, SDG 17 ("strengthen the means of implementation and revitalize the global partnership for sustainable development") still lags behind other SDGs [2].

Sustainability science is expected to play a fundamental role in implementing the SDGs. One aspect of this is that it should promote the adoption of a more ethical and equitable research culture that includes capacity building, knowledge-exchange activities, mutual trust, and respect between researchers from host nations and abroad [4]. Moreover, as sustainability pathways are regional and often country specific, sustainability science emphasizes the importance of regional and local contexts in the co-creation of scientific knowledge [10]. More balanced representation between North and South in sustainability research is crucial to shape how global sustainability challenges are defined, our relationships to these challenges, and how we think about studying them and designing global environmental policies [11]. However, to date evidence is lacking that sustainability science is leading the way in terms of ethical partnership between scientists worldwide.

This study examined whether ethical co-authoring is promoted in sustainability science articles through a quantitative analysis of the 300 articles published from the first issue of *Nature Sustainability* in January 2018 to February 2021. We used the journal *Nature* as a control to test whether research published under the banner of sustainability science favoured a more equitable involvement of authors from countries belonging to different income categories. We hypothesized that, given equal requirements for scientific excellence, *Nature Sustainability* articles should have a better representation of authors from LLMICs than the generalist journal *Nature*. Indeed, as the coproduction of inter- and transdisciplinary knowledge should be carried out through groups and truly collaborative networks, we expected that sustainability research put more emphasis on the equitable cooperation among LIC-LLMIC researchers than what is currently performed in "mainstream research" (i.e. as published in *Nature*). As inclusive authorship is only one criteria of scientific cooperation [12], we then broadened our analysis to explore gender disparity, the diversity of perspectives in different article categories, and authors' contributions to stages in knowledge production.

## Materials and methods

We retrieved all research articles published between 1 January 2018 and 25 February 2021 in HTML format from the *Nature Sustainability* website (https://www.nature.com/natsustain) through full access on 25 February 2021 (N = 300 articles). Our analysis was focused on the three article types that focus on original research: *Article* (a complex story often involving several techniques or approaches, N = 185), *Analysis* (a new analysis of existing data or new data obtained in a comparative analysis, N = 98) and *Brief communication* (a concise study with up

to 1,500 words, N = 17). The full text of each article was individually screened and classified in one of three categories depending on its geographical implementation (see **S1 Fig**): (i) articles without geographical consideration were categorized as 'Concept, modelling and technology' (N = 47), (ii) articles carried out in one specific country were categorized as 'Local studies' (N = 127), and (iii) articles referring to more than one country were categorized as 'International studies' (N = 126). In the 'Local study' articles, the focal country was identified by reading the full text. For comparative purposes, we also examined publications in *Nature* for the same period by retrieving all research articles in HTML format from the *Nature* website (https://www.nature.com/) through full access (N = 2994 articles).

The title, abstract, DOI, authors' affiliations, author contribution statements and funding information were automatically extracted from the HTML files using the 'rvest' R package for both *Nature* and *Nature Sustainability* publications. Authors' host countries were retrieved from their affiliations. When several affiliations were present, we considered the first affiliation only. Multiple institutional affiliations are a central concern in publication ethics [13]. There are both 'legitimate' multiple affiliations (i.e., where institutions substantially supported the study), and 'non-legitimate' multiple affiliations (where at least one of the affiliations is not reflecting a substantial contribution). We assumed that by selecting the first affiliation we minimized the risk of considering a non-legitimate affiliation. All countries–from both authors' affiliations and the location of 'Local study' papers–were harmonized using 'OpenRefine', and their ISO 3166–1 alpha-3 codes and geographical coordinates were retrieved from Wikidata. The country ISO 3166–1 alpha-3 codes were then used to determine the country's income group according to the World Bank classification based on 2019 gross national income. World Bank country categories were then used to analyse authorship patterns. For example, all articles for which at least one author was listed as having an LLMIC affiliation were classified as LLMIC articles, regardless of their position in the list of authors. All analyses were based on 'authorship events' (N = 2135) and not on author identity, meaning that the same author publishing several times appears several times in the analysis. For *Nature Sustainability* LLMIC authors (N = 80), we determined the gender of each author based on their name, completed with Internet searches of authors' profile with a photo. For LLMIC authors in *Nature* (N = 384), 29 had abbreviations as their first name and were excluded from the analysis. The remaining 355 authors were run through the genderizeR package [14]. GenderizeR provides a probability associated with the gender determined for a given name. We used only authors whose gender was determined with a probability > 0.75 (N = 303). Finally, we manually inspected authors' contributions in *Nature Sustainability* publications to determine their contributions to the different research stages in the CRediT author statement. To measure whether a more ethical co-authoring was promoted in *Nature sustainability* when compared to *Nature*, we calculated a difference in publication rate ($\Delta_{PR)}$) as follows:

$$\Delta_{PR} = \left. \frac{N_{country}}{N_{total}} - \frac{NS_{country}}{NS_{total}} \middle/ \frac{N_{country}}{N_{total}} + \frac{NS_{country}}{NS_{total}} \right.$$

where $N_{country}$ and $N_{total}$ are the number of articles in *Nature* for a given country and in total, respectively; and $NS_{country}$ and $NS_{total}$ are the number of articles in *Nature sustainability* for a given country and in total, respectively. $\Delta_{PR)}$ ranges between -1 (only *Nature Sustainability* articles) and 1 (only *Nature* articles).

## Results

The analysis revealed that the 300 papers in *Nature Sustainability* involved a total of 2135 authors, of which 80 (3.7%) came from a LLMIC and only nine (11.2%) were women. The

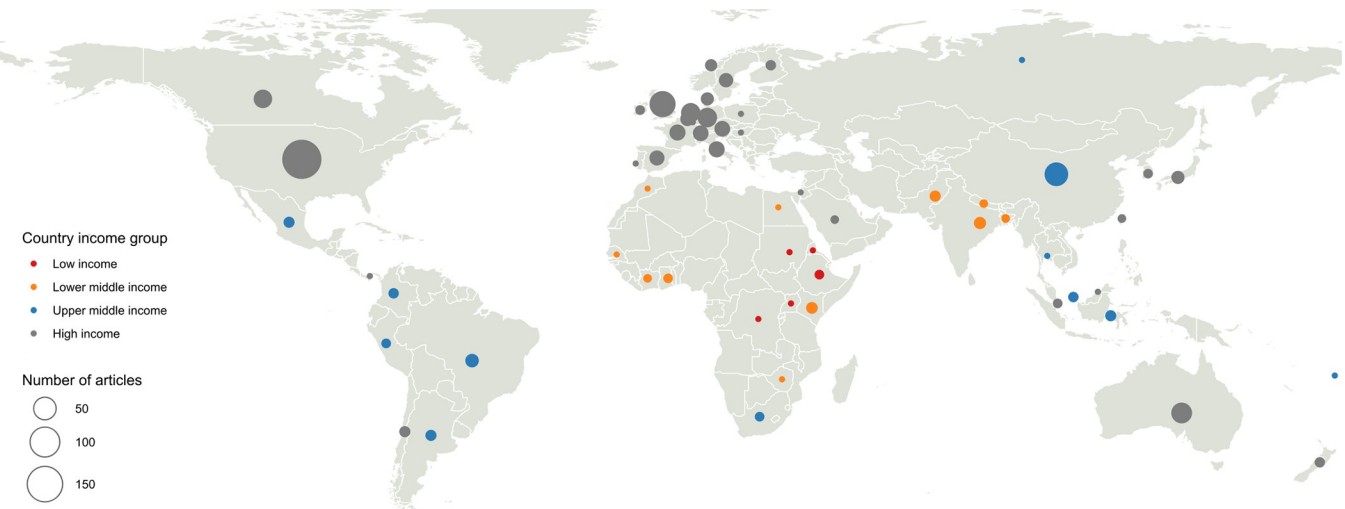

**Fig 1. World map of the number of publications in sustainability science between 2018 and 2021.** The number of articles is represented by circles of different sizes. The map was reprinted from https://www.naturalearthdata.com under a public domain license.

participation of LLMIC authors in *Nature Sustainability* was proportionally greater than in *Nature*: in this last journal, of the 2994 articles written by 46,817 authors over the same period, only 384 (0.8%) came from LLMICs, of which 71 (23.4%) were women. Of the 59 countries specified in author affiliations in *Nature Sustainability* articles, three accounted for more than half (59%) of authorship: the United States (34.8%), China (13,9%), and Great Britain (10.3%) (**Fig 1**). International co-authorships followed clear geographical lines from these countries, with the United States emerging as a hub (**Fig 2**).

The world map of collaboration confirmed the overall tendency of co-authors from LLMICs to be proportionally more involved in research published in *Nature Sustainability* than in *Nature* (more reddish dots in LLMICs; **Fig 3**). This was particularly clear for South American and Asia, while African countries showed a more mixed pattern. The overall smaller contribution of countries from LLMICs to *Nature* and *Nature Sustainability* publication reveals a pervasive lack of cooperation with HIC. Our analysis further revealed that entire regions such as French-speaking Africa appeared totally disconnected from the global network, with authorship representativeness similarly low in both journals. In *Nature Sustainability*, only 1.36% of co-authorship involved a French-speaking African country (Democratic Republic of Congo, Ivory Coast and Senegal). This corresponds to 0.04% and 0.6% of all publications in *Nature* (Ivory Coast and Senegal) and *Nature Sustainability* (Democratic Republic of Congo, Ivory Coast and Senegal), respectively.

We then deepened our analysis of *Nature Sustainability* articles to examine authorship with regard to publication category. We found that LLMIC authors were almost completely absent from the 47 articles regarding *Concept, methods and technologies*, and represented 4.3% of authors in *International studies* and 4.7% in *Local studies* (**Fig 4A**). While 48% of all publications concerned LLMICs, only 3.5% involved authors from these countries. In terms of the 28 *Local study* publications conducted in LLMICs, between 68.8% (lower medium income countries, LMIC) and 100% (lower income countries, LIC) involved at least one local researcher (**S2 Fig**). However these local researchers represented only 33% (LMIC) and 39% (LIC) of the 827 authors (**Fig 4B**). When further examining LLMIC authors' participation in these local studies, they were mainly involved in writing, investigation and data curation (**S3 Fig**). LLMIC institutions funded 7.1% of the local studies conducted in their own country.

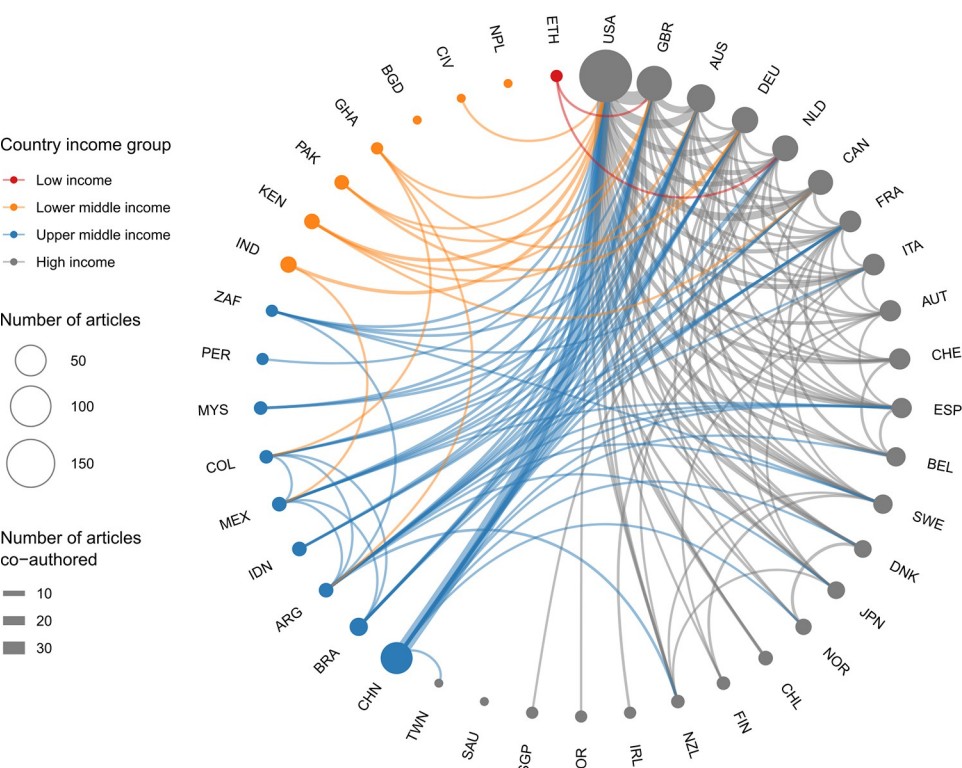

**Fig 2. Polar diagram of co-authorship among countries in *Nature Sustainability* between 2018 and 2021.** Lines indicate collaboration between countries, with line thickness indicating the intensity of the relationship (the greater the thickness, the greater the number of articles co-authored, and vice versa). Circle size is proportional to the number of articles and circle colours correspond to the country's income category. Line colours show the lowest income category between co-authoring countries. Lines for only one co-authorship are not shown for the purpose of clarity.

## Discussion

Transnational scientific knowledge that provides equal opportunities for every researcher represents a powerful way of moving towards sustainable development. In view of the UN 2030 Agenda, one would expect that sustainability science journals should act as path leaders in term of ethical partnership. However, there is no hard data in the literature about this assumption. By comparing authorship data between the journals *Nature sustainability* and *Nature*, our study intends to fill this gap.

Overall, these results show that High-Income Countries (HICs) glaringly shape global sustainability scientific production and discourse in *Nature Sustainability*, reflecting the North–South inequalities found in other journals [5, 6]. The causes of these inequalities are well documented; among others, lower tertiary education enrolment, scientific 'brain drain', lower research expenditure per capita, weaker institutional support and fewer funding opportunities [15]. While these causes have roots in LLMICs, in some cases back to their colonial history [16], HIC academia is yet to make significant progress in involving LLMICs to address the most urgent sustainability challenges. Over the last fifty years, North-South research collaborations has moved from HICs' researchers and funders assisting developing countries to find quick solutions to development issues to the building of local capacities in science and technology to formulating research partnership principles based on building mutual trust, learning and ownership [3]. However, in practice, these requirements are often not met for research and development projects. Scientific journals also have a role to play and can take concrete

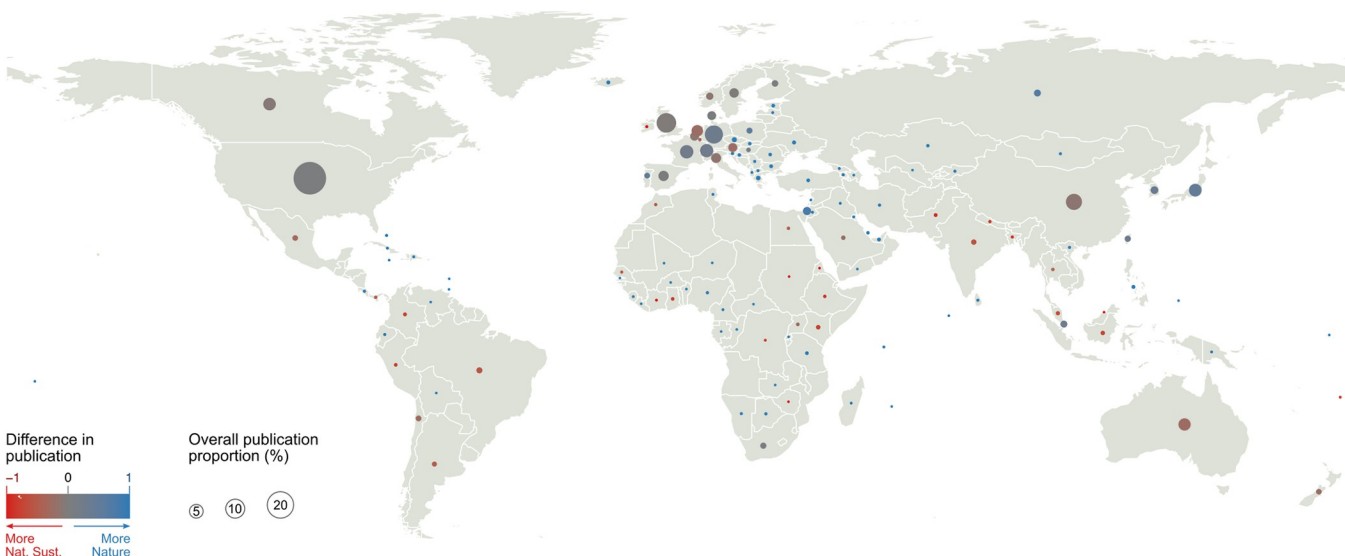

**Fig 3. Difference in publication rates between *Nature* and *Nature Sustainability*.** The colour scale indicates whether authors from the country publish more in *Nature* (blue) or *Nature Sustainability* (red) after accounting for total publication volume. Circle sizes reflect the country's proportion of publication combined across both journals. When several authors from the same country co-authored the same publication, the country has been considered only once. The calculation of the difference in publication score is given in the Methods. The map was reprinted from https://www.naturalearthdata.com under a public domain license.

actions to increase authorship diversity, including gender, and to recognize the contributions of host-nation researchers in the global South [17, 18]. The present manuscript was initially submitted to *Nature Sustainability* but the editorial decision was to reject it without review.

## a. Proportion of authorship across income groups

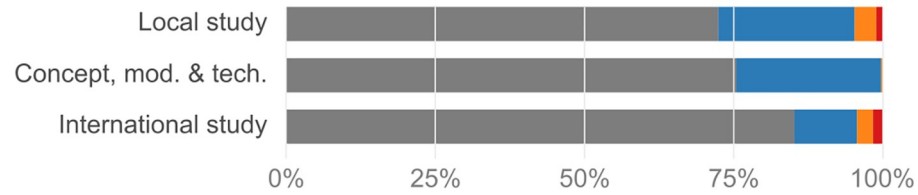

## b. Percentage of authors from the focal country

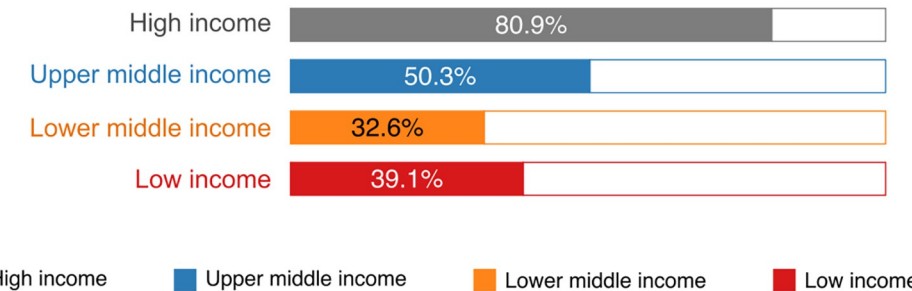

**Fig 4. Origin of co-authors of sustainability science articles. a**. Proportion of authorship across income groups for different types of publications in *Nature Sustainability*. **b**. Percentage of local authors among all co-authors involved in publications on a focal country (local studies).

Yet, we encourage this journal to develop a set of ethical partnership guidelines to attain a more balanced representation between North and South co-authors.

Nonetheless, our results also point to promising improvements in the involvement of LLMIC authors in top sustainability research articles. Proportionally, there were 4.6 times more LLMIC authors in *Nature Sustainability* than in *Nature* articles, and between 68.8% (LMIC) and 100% (LIC) of local studies were performed with in-country scientists (reflecting the discouragement of parachute research practices [5]) with the participation of local scientists in key research steps. By way of comparison, only 30% of articles in high-impact geoscience journals on an African topic contain an African author [6], while 60% of publications in coral reef biodiversity research performed in Indonesia and Philippines included host-nation scientists [5].

Of course, co-publication is an insufficient measure of partnership or collaboration strategies in research. Mutual capacity building and the translation of research results into policy interventions are increasingly seen as better indicators of successful North–South partnership [19] (e.g. SDG 17, indicator 6.1). Even the best studies may be of small benefit to local scientists and communities who need practicable solutions to face their specific issues, leading many universities in LLMICs to develop local initiatives to meet sustainability challenges [20]. Researchers, research institutions, scientific organizations, and funders from both the North and the South all have a role to play in transforming the current model of international collaboration, and there are some signs of improvement. Funding programmes such as the Belmont Forum (www.belmontforum.org) enforce equal research partnerships to create a better balance of power, require in-country research capacity building, and empower young LLMIC researchers to practice science in their country instead of supporting Northern researchers doing science in the South. The funding of ambitious researchers through Southern investment is another essential way of reducing publication inequality (e.g. the ARISE initiative in Africa, supported by the African Academy of Science and inspired by the ERC–European research council, www.ariseafrica.org). Another response to the widely acknowledged need to improve fairness in transnational collaborations is the increasing interest in the research fairness initiative–a self-reporting tool to identify strengths and weaknesses in research collaboration policy and practice (including data-sharing [21]) and to support the development of locally adapted research culture and infrastructure [22]. These ongoing initiatives should limit the foreign dependency of LLMICs and allow them to steer their own transformation agenda [23]. The inclusion of LLMICs in research so that studies 'come from inside' is the only way global academia can spur lasting change in sustainability science research and contribute to the 2030 Agenda.

## Supporting information

**S1 Fig. Procedure followed for the literature survey.**
(PDF)

**S2 Fig. Co-authorship of LLMIC authors in *Nature Sustainability Local study* publications. *Local study* articles are those carried out in one specific country (see Material and Methods).**
(PDF)

**S3 Fig. Contribution of LLMIC authors in *Nature Sustainability* publications.**
(PDF)

**S1 Data. Data set used for the study.**
(ZIP)

## Acknowledgments

We are grateful to the IRD staff for helping with the organisation of the meeting among the co-authors. We are grateful to three anonymous reviewers for their helpful comments on a previous version of the manuscript.

## Author Contributions

**Conceptualization:** Olivier Dangles.

**Formal analysis:** Olivier Dangles, Quentin Struelens.

**Software:** Quentin Struelens.

**Supervision:** Olivier Dangles.

**Visualization:** Quentin Struelens.

**Writing – original draft:** Olivier Dangles.

**Writing – review & editing:** Quentin Struelens, Mame-Penda Ba, Yvonne Bonzi-Coulibaly, Philippe Charvis, Evens Emmanuel, Carolina González Almario, Lahoucine Hanich, Ousmane Koita, Fabiola León-Velarde, Yvonne K. Mburu, Francine Ntoumi, Silvia Restrepo, Laurent Vidal.

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
