## [Decision Letter · Decision Letter 0]

24 May 2022

PONE-D-21-27967Insufficient yet improving involvement of the Global South in top sustainability science publicationsPLOS ONE

Dear Dr. Dangles,

Thank you for submitting your manuscript to PLOS ONE. After careful consideration, we feel that it has merit but does not fully meet PLOS ONE’s publication criteria as it currently stands. Therefore, we invite you to submit a revised version of the manuscript that addresses the points raised during the review process.

This paper is on a really important topic and I'm keen to see it published.   Reviewer 3 raises an important issue about why it was not published in a Nature family journal and I'd be keen to see the authors response on that.   There are number of other comments which require addressing, I'd be open to have you rebut a small number of these.

We look forward to receiving your revised manuscript.

Kind regards,

Alison Parker

Academic Editor

PLOS ONE

Journal Requirements:

4. We note that Figure 1 in your submission contain map images which may be copyrighted. All PLOS content is published under the Creative Commons Attribution License (CC BY 4.0), which means that the manuscript, images, and Supporting Information files will be freely available online, and any third party is permitted to access, download, copy, distribute, and use these materials in any way, even commercially, with proper attribution. For these reasons, we cannot publish previously copyrighted maps or satellite images created using proprietary data, such as Google software (Google Maps, Street View, and Earth). For more information, see our copyright guidelines: http://journals.plos.org/plosone/s/licenses-and-copyright.

Reviewers' comments:

Reviewer's Responses to Questions

**Comments to the Author**

1. Is the manuscript technically sound, and do the data support the conclusions?

Reviewer #1: No

Reviewer #2: Partly

Reviewer #3: Yes

2. Has the statistical analysis been performed appropriately and rigorously? 

Reviewer #1: Yes

Reviewer #2: No

Reviewer #3: I Don't Know

3. Have the authors made all data underlying the findings in their manuscript fully available?

Reviewer #1: Yes

Reviewer #2: Yes

Reviewer #3: No

4. Is the manuscript presented in an intelligible fashion and written in standard English?

Reviewer #1: Yes

Reviewer #2: No

Reviewer #3: Yes

5. Review Comments to the Author

Reviewer #1: Insufficient yet improving involvement of the Global South in top sustainability science publications

This article discusses a very important issue about involvement of Global South in top sustainability science publications. The materials and methods section is very well written. However, it is lacking in the data analysis part. I have the following comments in particular.

1. Line number 127: When the authors claim that the entire regions of French-speaking Africa appeared totally disconnected, they must support this claim with proper arguments and bring out the reasons for that disconnection.

2. Line number 157 to 159: In 68.8 – 100% study local researchers are involved, while 33-39% are given authorship. Does that mean the local researchers are not given authorship? If yes, what are the main reasons.

3. Discussion section: Most of the results do not follow from the study. The discussion part is a combination of literature only where the authors have cited many references. Therefore, the findings do not seem new and the reader wonders how this study is different. The authors are encouraged to do some more data analysis to distinguish their study from other studies. What is new that their study is contributing to the available literature is not very clear in the present manuscript.

Reviewer #2: 1) The paper concept is great and excellent contribution to existing body of knowledge

2) On aspect of data and it's presentation

i) Role of article type-good to examine article(Article , Analysis and Brief communication) across regions, or across gender

ii) Statistical analysis need to be improved ....any significant differences between the two journals studied?

3) Introduction-this is well written. On line 27, 'sister journal Nature'...can we avoid the term 'sister' (who descibed it a sister)?

Line 66....'examined this issue'....would be more appropriate to state the issue, to avoid word such as this issue. Paragraph 66-79 can be re-written to improve the flow.

3) Methodology

i) On materials and method section, key focus is on 'Nature Sustainability'. Nothing mentioned on Journal Nature.

ii) This need to be well elaborated a) Any other researcher working on such a study? Waht protocol did they use?

iii) Provide a brief description of the terms; Article , Analysis and Brief communication, either on introduction or methodology so they are clear to reader.

4) Results

Section can be improved to provide more inforamation to readers by designing the section into various sub-themes (sub-topics) eg i) gender effect....ii) Financing....iii) article type.. etc

5) Results can be improved. Line 118, use of words like; ' in that journal' should be avoided. May confuse readers as to which journal the term is reffering to.

6) On language used; This can be highly improved especially to avoid use of words such as; we (appearing severally especially on methods section), This study examined this issue (line 66), Yet to date evidence is lacking (line 63...reads like a continuation of previous line)

Reviewer #3: Very interesting and very important, if not a little limited in scope (2 journal comparison). However, the most glaring question is why not publish it in Nature Sustainability? It feels a little backhanded to publish about one journal in another. It would be interesting to know if the authors did submit it and have it rejected, and if so why? THAT would be an interesting part of the story. Asking Plos to publish this about another journal also creates a bit of an awkward friction between the two: is this going to start a heated exchange between the two? I would recommend not publishing this work in plos until it has been submitted and rejected by NS. That way, at least the process is clear.

Some further minor comments:

L25: not clear which journal “this” refers to

L26: I don’t agree with the formulation that “sustainability research” favoured equity--- it’s quite a logical leap. Frame this concept more concretely

L66: what does “this” in this issue refer to? What qualitative analysis was done? I see no evidence

L70: explain more fully and deeply this hypothesis: it’s the foundation of the paper and deserve more attention.

L98: discuss more fully the potential impacts of considering the first affiliation only and also your decision to do this. I don’t believe that the first affiliation is necessarily representative of anything but could be simply the order in which the affiliations were entered on the platform, or simply alphabetical.

L105: above you say there was no co-affiliations included, so now I’m confused

L109: how do you know what the gender of the name is?? Malawian names, for example, are often unisex: Chimwemwe can be a man or a woman. Even more Anglo-Saxon names like Leslie can be unisex. You just can’t know!!

L116: are these 9 from the total or from the LLMIC numbers?

L137: ratio of authors from where to where? Clarify

L158: I don’t understand how it can be 68-100%: are these min and max values? Confusing

L159: what do “these” refer to; again, 33-39% refers to what?

L160: “these” what?

L178: I would push the root cause argument back further to colonial history: it sounds like you’re blaming the poor countries. The root causes are well-documented!

L181: yes and no: journals can’t do much except reject papers that don’t meet certain criteria, but it is really the researchers and the funders who are to blame. More details on the concrete actions if this is your main issue, but I would like to see more people sharing the blame.

L186: again, this range is confusing

L198: provide a link/reference for this Belmont Forum; similarly with the ARISE initiative below

Figure 1A. The lines are too thin and too similar in colour to distinguish. Using blue on blue is really not good and the difference between orange and red is imperceptible. The dots in Europe are so covered with lines they are pointless. The legend “Number of authorship” doesn’t make sense- and is it for the dots or the lines? I really like the idea of this map but it needs to be totally redesigned and maybe rethought.

Figure 1B: similarly- the colours are so weak and indistinguishable: the outline on the circles is the same as the map colour which makes it even harder to see!

6. PLOS authors have the option to publish the peer review history of their article (what does this mean?). If published, this will include your full peer review and any attached files.

Reviewer #1: No

Reviewer #2: No

Reviewer #3: No

---

## [Author Response · Author response to Decision Letter 0]

21 Jun 2022

see attached files : cover_letter.doc and response_to_reviewers.doc

---

## [Decision Letter · Decision Letter 1]

22 Jul 2022

PONE-D-21-27967R1Insufficient yet improving involvement of the Global South in top sustainability science publicationsPLOS ONE

Dear Dr. Dangles,

Thank you for submitting your manuscript to PLOS ONE. After careful consideration, we feel that it has merit but does not fully meet PLOS ONE’s publication criteria as it currently stands. Therefore, we invite you to submit a revised version of the manuscript that addresses the points raised during the review process.

Just a couple of last comments from reviewer 3 to address whiuch shouldn't take long!

We look forward to receiving your revised manuscript.

Kind regards,

Alison Parker

Academic Editor

PLOS ONE

Journal Requirements:

Reviewers' comments:

Reviewer's Responses to Questions

**Comments to the Author**

1. If the authors have adequately addressed your comments raised in a previous round of review and you feel that this manuscript is now acceptable for publication, you may indicate that here to bypass the “Comments to the Author” section, enter your conflict of interest statement in the “Confidential to Editor” section, and submit your "Accept" recommendation.

Reviewer #1: All comments have been addressed

Reviewer #3: All comments have been addressed

2. Is the manuscript technically sound, and do the data support the conclusions?

Reviewer #1: Yes

Reviewer #3: Yes

3. Has the statistical analysis been performed appropriately and rigorously? 

Reviewer #1: Yes

Reviewer #3: Yes

4. Have the authors made all data underlying the findings in their manuscript fully available?

Reviewer #1: Yes

Reviewer #3: Yes

5. Is the manuscript presented in an intelligible fashion and written in standard English?

Reviewer #1: Yes

Reviewer #3: Yes

6. Review Comments to the Author

Reviewer #1: The authors have adequately addressed my comments raised in a previous round of review. I recommend to accept.

Reviewer #3: I still think it's important to mention IN THE MANSCRIPT, not just to the reviewers, that you submitted and were rejected from Nature Sustainability. It will be a persistent question if it is not addressed up front. Also, there is a mix of italics and non italics in the namin of the 2 journals.

7. PLOS authors have the option to publish the peer review history of their article (what does this mean?). If published, this will include your full peer review and any attached files.

Reviewer #1: No

Reviewer #3: **Yes: **Elizabeth Tilley

---

## [Author Response · Author response to Decision Letter 1]

30 Jul 2022

As requested, we have now mentioned in the discussion on the manuscript that we submitted and were rejected from Nature Sustainability. We have also put in italics the naming of the 2 journals.

L228: " The present manuscript was initially submitted to Nature Sustainability but the editorial decision was to reject it without review. Yet, we encourage this journal to develop a set of ethical partnership guidelines to attain a more balanced representation between North and South co-authors."

---

## [Editor Report · Decision Letter 2]

3 Aug 2022

Insufficient yet improving involvement of the Global South in top sustainability science publications

PONE-D-21-27967R2

Dear Dr. Dangles,

We’re pleased to inform you that your manuscript has been judged scientifically suitable for publication and will be formally accepted for publication once it meets all outstanding technical requirements.

Kind regards,

Alison Parker

Academic Editor

PLOS ONE
---

## [Editor Report · Acceptance letter]

23 Aug 2022

PONE-D-21-27967R2 

Insufficient yet improving involvement of the Global South in top sustainability science publications 

Dear Dr. Dangles:

I'm pleased to inform you that your manuscript has been deemed suitable for publication in PLOS ONE. Congratulations! Your manuscript is now with our production department. 

Kind regards, 

on behalf of

Dr. Alison Parker 

Academic Editor

PLOS ONE